# Are degree of urbanisation and travel times to healthcare services associated with the processes of care and outcomes of heart failure? A retrospective cohort study based on administrative data

Jacopo Lenzi[1], Vera Maria Avaldi [1,2]*, Dario Molinazzi[3], Carlo Descovich[2], Stefano Urbinati[4], Veronica Cappelli[5], Maria Pia Fantini[1]

**1** Department of Biomedical and Neuromotor Sciences, Alma Mater Studiorum – University of Bologna, Bologna, Italy, **2** Department of Clinical Governance and Quality, Bologna Local Healthcare Authority, Bologna, Italy, **3** Department of Management Control and Administrative Data, Bologna Local Healthcare Authority, Bologna, Italy, **4** Department of Cardiology, Bellaria Hospital, Bologna, Italy, **5** Directorate of Assistance, Technology and Rehabilitation, Bologna Local Healthcare Authority, Bologna, Italy

* veramaria.avaldi@ausl.bologna.it

## Abstract

A few studies have found that patients with heart failure (HF) living in less densely populated areas have reduced use of services and poorer outcomes. However, there is a lack of evidence regarding transport accessibility measured as the actual distance between the patient's home and the healthcare facility. The aim of this study was to investigate if different urbanisation levels and travel times to healthcare services are associated with the processes of care and the outcomes of HF. This retrospective cohort study included patients residing in the Local Healthcare Authority of Bologna (2915 square kilometres) who were discharged from hospital with a diagnosis of HF between 1 January and 31 December 2017. Six-month study outcomes included both process (cardiology follow-up visits) and outcome measures (all-cause readmissions, emergency room visits, all-cause mortality). Of the 2022 study patients, 963 (47.6%) lived in urban areas, 639 (31.6%) in intermediate density areas, and 420 (20.8%) in rural communities. Most patients lived ≤30 minutes away from the nearest healthcare facility, either inpatient or outpatient. After controlling for a number of individual factors, no significant association between travel times and outcomes was present. However, rural patients as opposed to urban patients were more likely to see a cardiologist during follow-up (OR 1.42, 99% CI 1.03–1.96). These follow-up visits were associated with reduced mortality within 6 months of discharge (OR 0.53, 99% CI 0.32–0.87). We also found that multidisciplinary interventions for HF were more common in rural than in urban settings (18.8% vs. 4.0%). In conclusion, travel times had no impact on the quality of care for patients with HF. Differences between urban and rural patients were possibly mediated by more proximal factors, some of which are potential targets for intervention such as the availability and utilisation of follow-up cardiology services and multidisciplinary models of care.

**Data Availability Statement:** All relevant data are within the manuscript and its Supporting Information files.

**Funding:** The authors received no specific funding for this work.

**Competing interests:** The authors have declared that no competing interests exist.

# Introduction

Heart failure (HF) is a complex clinical syndrome with a prevalence of 1 to 2% in the adult population of Western countries, a value that exceeds 10% in individuals over 70 years of age [1,2]. The prognosis is poor, with an in-hospital mortality of 10% and a one-year mortality after discharge of 20 to 40% [3–8]. HF is one of the main reasons for hospital admission in Europe and the US [9], especially for patients over 65 years [10], and the readmission rate exceeds 20% and 50% within a month and a year of hospital discharge, respectively [11,12]. Patients require a care approach that takes into account all their complex care needs [13], and indeed it has been shown that easy access to care is a significant determinant of their outcomes [14].

A recent systematic review regarding accessibility in terms of distance and travel time from the patient's home, showed that individuals living near healthcare facilities have better health outcomes or a higher rate of access to services than those living further away [15]. However, this review included mainly cancer research studies relying on different data sources and variables.

With respect to HF, a few studies found that living in less densely populated areas was associated with reduced use of services and poorer outcomes [16–18]. These findings should be read keeping in mind that geographic position might have a close correlation with a large number of variables, such as socioeconomic status (SES), availability of appropriate services, and even peculiar clinical conditions. A potential limitation of these studies is the lack of information regarding transport accessibility measured as the actual distance between the patient's home and the healthcare facility.

The Local Healthcare Authority (LHA) of Bologna, located in the Emilia-Romagna region of Italy, has various hospitals and primary care services in its territory. Over the past few years, an increasing number of healthcare homes and ambulatory care nursing practices have been established, where patients are medically examined and receive outpatient nursing interventions for lifestyle change or for the control and management of their own health problems. Many of the interventions included in the care pathways (CPs), such as the heart failure CP (HF-CP), are also provided within these care settings. This multidisciplinary and standardised CP promotes the integration of different services and professionals, and aims to improve the health status of the patients with HF through a direct and easy access to care.

Despite the large number of healthcare facilities and the efforts to improve patients' access to services, the catchment area of Bologna has a relatively diverse geography and uneven population distribution. Therefore, the objective of this study was to investigate whether different travel times to healthcare services in the LHA of Bologna are associated with the processes of care and the outcomes of patients with HF. In keeping with existing literature, we also evaluated the impact of urbanisation levels on the study processes and outcomes.

# Materials and methods

## Setting and study population

This retrospective observational study included all hospital discharges in the LHA of Bologna with a primary diagnosis of HF (ICD-9-CM codes: 398.91, 402.x1, 404.x1, 404.x3, 428.xx) between 1 January and 31 December 2017. The LHA of Bologna is located in the northeast of Italy, has a population of about 876,000 and covers an area of 2915 km$^2$, with a territory that is 29% mountains (the Apennines), 32% hills, and 39% plains (the Po Valley) (Figure A in S1 Fig).

Data were retrieved from the Hospital Discharge Records (HDRs) Database (this and the other data sources used in this study are described in S1 Table) [19]. For patients with multiple eligible hospital admissions over the one-year study period, we considered the first one as the index admission. Repeated admissions within one day of discharge were regarded as one single episode of care, and the beginning of the follow-up was set at the discharge date of the episode of care. All patients were followed up to 6 months.

Patients were excluded if any of the following criteria were met (S2 Fig):

1. Permanent and/or current address outside the LHA catchment area

2. Homeless

3. Registered at a general practitioner (GP) who practiced outside the catchment area

4. Age >100 years, because very old patients may have distinctive clinical features at diagnosis and survival

5. Planned hospital admission, to focus analyses on acute/urgent episodes of care

6. Transfer from another facility, to focus analyses on incident cases of HF

7. Daytime hospital care, i.e., one-day admissions to the hospital without overnight stay to perform diagnostic procedures and/or surgical, therapeutic or rehabilitative care

8. A secondary diagnosis of non-cardiogenic acute pulmonary oedema (ICD-9-CM 518.4), i.e., patients with symptoms probably related to causes other than HF

9. A secondary diagnosis of acute kidney failure (ICD-9-CM 584.x), i.e., patients whose reason for hospitalisation is likely not to be HF

10. Pregnancy, childbirth or puerperium (Major Diagnostic Category 14)

11. A major procedure on the cardiovascular system (ICD-9-CM 00.5x, 00.66, 35.xx, 36.xx, 37.31–37.66, 37.70–37.89, 37.94–37.98), i.e., patients with severe cardiac impairment as the main reason for hospitalization

12. Death during the index episode of care

13. Discharge against medical advice

14. Length of stay >90 days, i.e., very complex or unstable cases

15. Access to residential care facility for the elderly before index hospitalisation or during follow-up, i.e., patients being given end-of-life care.

## Study outcomes

The 6-month outcomes evaluated in this study included both process and outcome measures:

1. Cardiology follow-up visits provided in either outpatient or inpatient cardiology services, except for those booked prior the index admission or performed during hospital stays (source: Outpatient Care Database [OCD])

2. All-cause unplanned readmissions occurred at any hospital within 2 to 180 days of discharge, and lasting >1 day (source: HDRs)

3. Emergency room (ER) visits not related to injuries and not resulting in inpatient admission (source: ER database)

4. All-cause mortality (source: vital registration system).

## Degree of urbanisation

Using the Eurostat's Degree of Urbanisation (DEGURBA) classification system (revised definition, 2014), the 45 municipalities (*comuni*) where the patients lived were subdivided into rural areas (alternative name: sparsely populated areas), towns or suburbs (intermediate density areas), and cities (densely populated areas). As illustrated in Figure B in S1 Fig, the city of Bologna was classified as urban (388,000 pop., 44%), the nearby *comuni* and other areas in the Po Valley were classified as towns or suburbs (304,000 pop., 35%), while the remaining *comuni*—both flat and mountainous—were classified as rural (184,000 pop., 21%).

## Travel times to healthcare services

Because of the catchment area's diverse geography, we calculated the travel times between the patients' home addresses (source: civil registry) and a series of healthcare facilities that provide care for patients with HF. These included:

1. Emergency rooms (*N* = 12)

2. Cardiology wards (*N* = 5)

3. Outpatient cardiology services (*N* = 27)

4. Ambulatory care nursing practices (*N* = 34), 15 of which are located in the healthcare homes

5. GP practices (*N* = 739), run by a total of 525 GPs.

Due to proximity to the border, three hospitals providing emergency and/or cardiology care outside the LHA catchment area were included in the study.

After geocoding all locations in the WGS84 spatial reference by means of the Stata `opencagegeo` package, the travel times between geographic coordinates were computed using the `georoute` package [20,21]. `georoute` calculates how long it takes to drive the distance between two points under average traffic conditions [21].

To account for potential nonlinear relationships with the outcomes, travel times were split into five categories: ≤5 min (very short), >5–10 min (short), >10–20 min (medium), >20–30 min (long), and >30 min (very long).

In addition to the degree of urbanisation, different sets of routing distances were considered as the potential predictors of each study outcome. When we analysed cardiology follow-up visits, the travel time of interest was to the nearest cardiology service, either inpatient or outpatient. In all the other analyses, we considered three distinct travel times: to the nearest ER, the nearest patient's GP practice, and the nearest outpatient service, either cardiologist or non-cardiologist. Moreover, follow-up cardiology visits were treated as potential predictors of hospital readmissions, ER visits and mortality [22–24]. An overview of these sets of variables is provided in S2 Table.

## Potential confounders

We collected some patient baseline characteristics to reduce the potential source of confounding. These included:

1. Age

2. Sex

3. Citizenship

4. Length of stay

5. Provision of intensive care during hospital stay

6. Discharge from a cardiology ward

7. Thirty-one Elixhauser conditions identified in the index episode of care and in all hospital admissions occurring two-years prior to the index hospitalisation [25], plus four additional conditions not included in the Elixhauser's list (myocardial infarction [ICD-9-CM 410.x, 412], cerebrovascular diseases [362.34, 430.x–438.x], dementia [290.x, 294.1, 331.2], leukaemia [204.x-208.x])

8. Use of 10 drug therapies one-year prior to the index admission (≥1 filled prescription) (source: Outpatient Pharmaceutical Database).

See Tables A and B in S3 Table for the detailed list of drug therapies and Elixhauser comorbidities. We also took into account the following information in the analyses [19,26–29]:

1. Use of first-line medications during follow-up, that is, angiotensin-converting enzyme inhibitors/angiotensin receptor blockers (ACEIs/ARBs) and β-blockers (only for hospital readmission, ER visit and mortality analyses)

2. Registration at occasional or full-time general home-care services as a proxy of social and medical complexity

3. Registration at the HF-CP, a structured multidisciplinary care plan that promotes integration between primary and secondary care, and details essential steps in the care of patients. The GP remains the gatekeeper for the patients and coordinates with cardiologists and nurses for an easier access to consultation and counselling to improve lifestyle and optimise medication adherence. The HF-CP can involve either outpatient clinic-based interventions (clinic-based HF-CP) or home visits (home-based HF-CP), depending on the patient's clinical condition and ability to move. There are no facilities specifically dedicated to the HF-CP: registered patients can access GP practices, cardiology services or ambulatory care nursing practices in case they need medical care or counselling.

The patient could already be registered at the beginning of the follow-up, or access the outpatient services following discharge. Information on outpatient care was collected from regional and LHA administrative databases. All the data sources used in this study are de-identified and linkable using the unique patient identifier.

As previously mentioned, cardiology visits during follow-up were included in regression analyses as potential predictors of hospital readmission, ER visits and mortality [19,22–24].

## Statistical analysis

Continuous variables were summarised as mean ± standard deviation; discrete and categorical variables were summarised as frequencies and percentages. Comparisons across urbanisation levels were performed using one-way analysis of variance, Kruskal-Wallis test or chi-squared test, when appropriate. The spatial distribution of homes and healthcare facilities was graphically displayed with the aid of dot maps.

To ensure an equal time window for detecting and measuring time-varying covariates, such as outpatient care and medication use (see section above), the impact of urbanisation level and

travel times on the study outcomes was assessed using a time-matched nested case-control design. Patients who experienced the study outcome were defined as cases, and 9 controls were randomly selected and matched to each case by gender, age group (defined using a decile split) and follow-up duration. This technique is called "incidence density sampling". Odds ratios (ORs) were estimated by conditional logistic regression models to account for the matching of cases and controls [19].

All regression models included the potential confounders described earlier. However, to avoid overfitting and misclassification, not all comorbidities and previous drug therapies were included in the models. A subset of all candidate variables was preliminary chosen for inclusion using an automated selection method which is described in detail elsewhere [19,30]. In brief, a bootstrap procedure was adopted to determine which comorbidities were significantly associated with the outcomes. Using this approach, a backward elimination of potential confounders was applied in each replicated sample with a significance level or removal equal to 0.05, and only risk factors selected in at least 50% of the replicates were included as confounders in the final multivariable regression models. The confounders included in the final models are reported in table footnotes. The variance inflation factor, a measure of correlation among predictor variables (multicollinearity), was <3 for all of the predictors included in the models.

To control for type I error related to multiple testing, the significance level was set at 0.01. All analyses were carried out using Stata software, version 15 (StataCorp. 2017. *Stata Statistical Software*: *Release 15*. College Station, TX: StataCorp LLC).

### Ethics statement

Ethical approval to undertake this research was granted from the *Comitato Etico di Area Vasta Emilia Centro* (Submission Number 254/2019/OSS/AUSLBO).

This retrospective study was carried out in conformity with the regulations on data management with the Italian law on privacy (Legislation Decree 196/2003 amended by Legislation Decree 101/2018). Data were pseudonymised prior to the analysis at the regional statistical office, and each patient was assigned a unique identifier that eliminates the ability to trace the patient's identity or other sensitive data. Pseudonymised administrative data can be used without a specific written informed consent when patient information is collected for healthcare management and healthcare quality evaluation and improvement (according to art. 110 on medical and biomedical and epidemiological research, Legislation Decree 101/2018).

Patients and the public were not involved in the design or planning of the study. All procedures performed in this study were in accordance with the 1964 Helsinki Declaration and its later amendments.

### Results

Of the 3138 patients discharged after HF, 2022 (64.4%) met the inclusion criteria (S2 Fig). Mean age was 82.2 ± 9.5 years and 1089 (53.9%) were females. A total of 963 (47.6%) patients lived in densely populated areas, 639 (31.6%) lived in intermediate density areas and 420 (20.8%) lived in rural areas. As shown in Table 1, patients living in rural areas were on average younger, more often registered at the HF-CP and more often discharged from internal medicine services. Specific comorbidities and previous drug therapies are summarised in Tables A and B in S4 Table.

Travel times are summarised in Table 1, while the spatial distribution of HF patients and healthcare services in the LHA catchment area is illustrated in Fig 1. Although outpatient healthcare services were relatively scattered as compared to inpatient services, most patients lived ≤30 minutes away from the nearest facility. Still, we found that travel times were

**Table 1. Distribution of patient characteristics and travel times, overall and by degree of urbanisation.**

| Patient characteristics | All | | City | | Towns and suburbs | | Rural areas | | P |
|---|---|---|---|---|---|---|---|---|---|
| | (n = 2022) | | (n = 963) | | (n = 639) | | (n = 420) | | |
| | n | % | n | % | n | % | n | % | |
| Females | 1089 | 53.9 | 537 | 55.8 | 340 | 53.2 | 212 | 50.5 | 0.178 |
| Age, mean ± SD | 82.2 ± 9.5 | | 82.8 ± 9.6 | | 82.1 ± 8.9 | | 80.9 ± 10.0 | | 0.002 |
| Non-Italians | 35 | 1.7 | 21 | 2.2 | 9 | 1.4 | 5 | 1.2 | 0.324 |
| Length of stay, mean ± SD | 8.0 ± 4.8 | | 9.3 ± 7.4 | | 8.9 ± 7.6 | | 9.3 ± 7.1 | | 0.686 |
| Two or more comorbidities | 1705 | 84.3 | 795 | 82.6 | 539 | 84.4 | 371 | 88.3 | 0.060 |
| Two or more previous drug therapies | 1729 | 85.5 | 807 | 83.8 | 554 | 86.7 | 368 | 87.6 | 0.212 |
| Discipline of the ward of discharge | | | | | | | | | <0.001 |
| Internal medicine | 1436 | 71.0 | 645 | 67.0 | 449 | 70.3 | 342 | 81.4 | |
| Geriatrics | 285 | 14.1 | 165 | 17.1 | 84 | 13.1 | 36 | 8.6 | |
| Cardiology | 272 | 13.5 | 136 | 14.1 | 97 | 15.2 | 39 | 9.3 | |
| Other | 29 | 1.4 | 17 | 1.8 | 9 | 1.4 | 3 | 0.7 | |
| Intensive care | 67 | 3.3 | 27 | 2.8 | 22 | 3.4 | 18 | 4.3 | 0.358 |
| Use of ACEIs/ARBs | 1042 | 51.5 | 499 | 51.8 | 334 | 52.3 | 209 | 49.8 | 0.706 |
| Use of β-blockers | 1457 | 72.1 | 671 | 69.7 | 461 | 72.1 | 325 | 77.4 | 0.013 |
| Use of both ACEIs/ARBs and β-blockers | 817 | 40.4 | 384 | 39.9 | 263 | 41.2 | 170 | 40.5 | 0.877 |
| HF care pathway | | | | | | | | | <0.001 |
| No | 1817 | 89.9 | 924 | 96.0 | 552 | 86.4 | 341 | 81.2 | |
| Clinic-based | 115 | 5.7 | 31 | 3.2 | 35 | 5.5 | 49 | 11.7 | |
| Home-based | 90 | 4.5 | 8 | 0.8 | 52 | 8.1 | 30 | 7.1 | |
| General home care | | | | | | | | | 0.635 |
| No | 497 | 51.6 | 310 | 48.5 | 221 | 52.6 | 1028 | 50.8 | |
| Occasional | 50 | 5.2 | 38 | 5.9 | 20 | 4.8 | 108 | 5.3 | |
| Full-time | 416 | 43.2 | 291 | 45.5 | 179 | 42.6 | 886 | 43.8 | |
| Travel time to nearest ER | | | | | | | | | <0.001 |
| Very short (≤5 min) | 199 | 9.84 | 118 | 12.30 | 23 | 3.60 | 58 | 13.81 | |
| Short (>5–10 min) | 755 | 37.34 | 622 | 64.59 | 72 | 11.27 | 61 | 14.52 | |
| Medium (>10–20 min) | 837 | 41.39 | 221 | 22.90 | 441 | 69.01 | 175 | 41.67 | |
| Long (>20–30 min) | 191 | 9.40 | 2 | 0.21 | 101 | 15.81 | 88 | 21.00 | |
| Very long (>30 min) | 40 | 1.98 | 0 | 0.00 | 2 | 0.31 | 38 | 9.00 | |
| Travel time to nearest patient's GP practice | | | | | | | | | 0.001 |
| Very short (≤5 min) | 1,328 | 65.7 | 634 | 65.8 | 440 | 68.9 | 254 | 60.5 | |
| Short (>5–10 min) | 424 | 21.0 | 227 | 23.6 | 120 | 18.8 | 77 | 18.3 | |
| Medium (>10–20 min) | 215 | 10.6 | 97 | 10.1 | 59 | 9.2 | 59 | 14.0 | |
| Long (>20–30 min) | 35 | 1.7 | 2 | 0.2 | 17 | 2.7 | 16 | 3.8 | |
| Very long (>30 min) | 20 | 1.0 | 3 | 0.3 | 3 | 0.5 | 14 | 3.3 | |
| Travel time to nearest cardiology service, either inpatient or outpatient[a] | | | | | | | | | <0.001 |
| Very short (≤5 min) | 865 | 42.8 | 478 | 49.6 | 291 | 45.5 | 96 | 22.9 | |
| Short (>5–10 min) | 775 | 38.3 | 468 | 48.6 | 195 | 30.5 | 112 | 26.7 | |
| Medium (>10–20 min) | 270 | 13.4 | 17 | 1.8 | 122 | 19.1 | 131 | 31.2 | |
| Long (>20–30 min) | 87 | 4.3 | 0 | 0.0 | 31 | 4.9 | 56 | 13.3 | |
| Very long (>30 min) | 25 | 1.2 | 0 | 0.0 | 0 | 0.0 | 25 | 6.0 | |
| Travel time to nearest ambulatory care nursing practice | | | | | | | | | <0.001 |
| Very short (≤5 min) | 1040 | 51.4 | 553 | 57.4 | 371 | 58.1 | 116 | 27.6 | |
| Short (>5–10 min) | 734 | 36.3 | 401 | 41.6 | 207 | 32.4 | 126 | 30.0 | |

*(Continued)*

**Table 1.** (Continued)

| Patient characteristics | All | | City | | Towns and suburbs | | Rural areas | | P |
|---|---|---|---|---|---|---|---|---|---|
| | (n = 2022) | | (n = 963) | | (n = 639) | | (n = 420) | | |
| | n | % | n | % | n | % | n | % | |
| Medium (>10–20 min) | 209 | 10.3 | 9 | 0.9 | 61 | 9.5 | 139 | 33.1 | |
| Long (>20–30 min) | 39 | 1.9 | 0 | 0.0 | 0 | 0.0 | 39 | 9.3 | |
| Very long (>30 min) | 0 | 0.0 | 0 | 0.0 | 0 | 0.0 | 0 | 0.0 | |

SD, standard deviation; ACEIs/ARBs, angiotensin-converting enzyme inhibitors/angiotensin receptor blockers; HF, heart failure; ER, emergency room; GP, general practitioner.

[a] Nearest cardiology ward or outpatient cardiology service.

significantly dependent on urbanisation, as patients living in rural communities were more distant from all healthcare services than those living in more densely populated areas.

The outcomes rates over the 6-month observation period, overall and by degree of urbanisation, are reported in Table 2. There was a lower rate of cardiology visits among patients living in Bologna. No other crude associations between urbanisation level and outcomes were found.

The impact of urbanisation levels and travel times resulting from multivariable regression analysis is presented in Tables 3 and 4. After adjusting for travel times and other patient characteristics (see table footnotes), we found that rural patients were more likely to see a cardiologist (OR 1.42, 99% CI 1.03–1.96), compared with urban patients. No other significant association between predictors and outcomes was present.

## Other outcome predictors

The full multivariable regression models including travel times and processes of care, both inpatient and outpatient, are presented in Tables A and B in S5 Table. The main results can be summarised as follows:

1. Patients registered at the clinic-based HF-CP were more likely to be seen by a cardiologist during follow-up (OR 1.69, 99% CI 1.02–2.80)

2. As compared with patients with no cardiology follow-up visits, patients seen by a cardiologist during follow-up were less likely to die within 6 months of discharge (OR 0.53, 99% CI 0.32–0.87).

## Discussion

The main result of this retrospective cohort study was that urbanisation and travel times to healthcare services had no impact on the processes of care and the outcomes of patients with HF in the 6 months following hospital discharge. The only exception was that cardiology follow-up visits were more frequent among rural than among urban patients. We also found that patients registered at the clinic-based HF-CP were more likely to see a cardiologist during follow-up, and that outpatient cardiology care was associated with improved outcomes.

These results were obtained after controlling for several potential individual confounders. However, our administrative databases do not include very relevant clinical information, such as left ventricular ejection fraction and classification of disease severity, which can affect the patient outcomes.

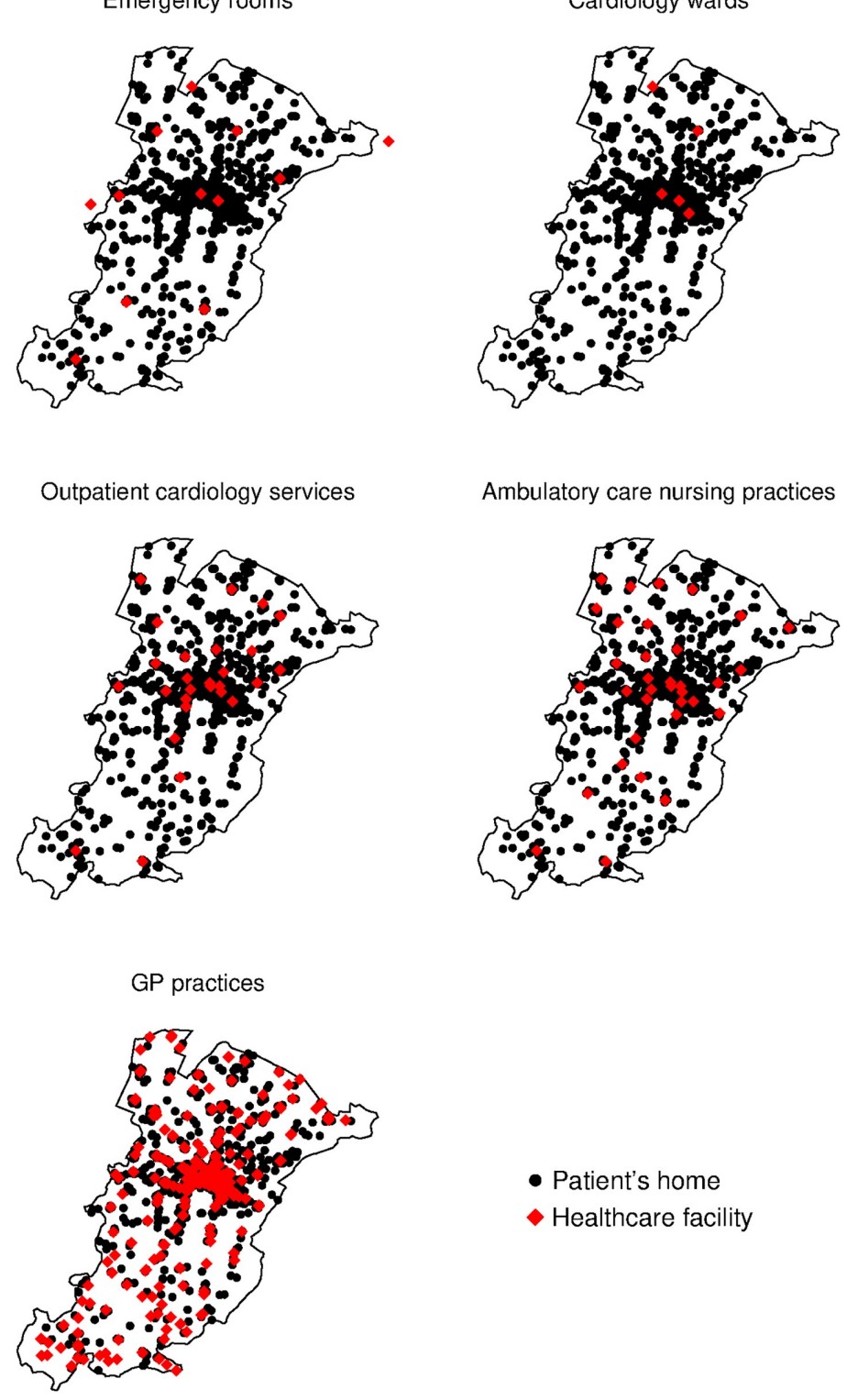

**Fig 1. Dot map of patient's homes and inpatient/outpatient healthcare services, Local Healthcare Authority of Bologna, year 2017.** The shapefiles used to make this figure are made publicly available by the Italian National Institute of Statistics. Administrative boundaries reproduced from [31] under a CC BY license, with permission from the Italian National Institute of Statistics, original copyright 2019.

**Table 2. Outcomes rates (%) at 6 months after heart failure discharge, overall and by degree of urbanisation.**

| Six-month study outcome | All | | City | | Towns and suburbs | | Rural areas | | P |
|---|---|---|---|---|---|---|---|---|---|
| | (n = 2022) | | (n = 963) | | (n = 639) | | (n = 420) | | |
| | n | % | n | % | n | % | n | % | |
| Cardiology follow-up visit[a] | 649 | 32.1 | 272 | 28.2 | 224 | 35.1 | 153 | 36.4 | 0.002 |
| All-cause unplanned readmission | 726 | 35.9 | 357 | 37.1 | 213 | 33.3 | 156 | 37.1 | 0.261 |
| ER visit not resulting in inpatient admission[b] | 487 | 24.1 | 213 | 22.1 | 168 | 26.3 | 106 | 25.2 | 0.132 |
| All-cause mortality | 391 | 19.3 | 190 | 19.7 | 119 | 18.6 | 82 | 19.5 | 0.855 |

[a] Of the 649 patients with cardiology follow-up visits, 91 (14.0%) were seen within 14 days of discharge, 195 (30.0%) within 28 days and 484 (74.6%) within 90 days.

[b] Of the 487 patients seeking emergency care, 362 (74.3%) accessed the nearest ER.

To the best of our knowledge, this is the first study to evaluate whether the care of patients with HF is associated with transport accessibility to healthcare, measured as the travel time to a number of different services and facilities. A strength of our method is that the patient's home address was the starting point to measure the travel times, while a potential limitation is that these calculations were made assuming that all patients would attend the nearest facility. Another strength is that our estimates of driving times were not only a function of driving distances [15,32], but also depended on average traffic volumes, road conditions and elevation. Nevertheless, travel time is only a component of the travel burden for these patients, who are elderly and can face a relevant combination of barriers due to disabilities, clinical conditions and comorbidities, and need for multiple medical assessments [33]. Also, often elderly patients must be accompanied by their caregivers, whose availability depends, among other things, on the opportunity to take some time off work or to delegate the care of children. This important aspect would deserve further investigation, but administrative databases do not provide any information on informal caregivers.

**Table 3. Impact of urbanisation level and travel time to the nearest cardiology service on the likelihood of seeing a cardiologist within 6 months of heart failure discharge.**

| Urbanisation and travel times | Cardiology follow-up visit | |
|---|---|---|
| | OR[a] | 99% CI |
| Degree of urbanisation | | |
| City | 1.00 | |
| Towns or suburbs | 1.17 | 0.90–1.52 |
| Rural area | 1.42[b] | 1.03–1.96 |
| Travel time to nearest in- or outpatient cardiology service | | |
| Very short (≤5 min) | 1.00 | |
| Short (>5–10 min) | 0.86 | 0.67–1.10 |
| Medium (>10–20 min) | 0.95 | 0.68–1.34 |
| Long (>20–30 min) | 0.63 | 0.35–1.14 |
| Very long (>30 min) | 0.54 | 0.17–1.67 |

OR, odds ratio; CI, confidence interval.

[a] Adjusted for age, sex, length of stay, intensive care, discharge from cardiology, dementia, and general home care/HF care pathway during follow-up.

[b] Significant at the 0.01 level.

**Table 4. Impact of urbanisation level and travel times to healthcare services on hospital readmissions, emergency room visits and mortality within 6 months of heart failure discharge.**

| Urbanisation and Travel Times | All-Cause Unplanned Readmission | | ER Visit | | All-Cause Mortality | |
|---|---|---|---|---|---|---|
| | OR[a] | 99% CI | OR[b] | 99% CI | OR[c] | 99% CI |
| Degree of urbanisation | | | | | | |
| City | 1.00 | | 1.00 | | 1.00 | |
| Towns or suburbs | 0.77 | 0.57–1.04 | 1.10 | 0.37–2.03 | 0.96 | 0.63–1.48 |
| Rural area | 0.85 | 0.60–1.19 | 1.26 | 0.95–1.67 | 0.96 | 0.57–1.53 |
| Travel time to nearest ER | | | | | | |
| Very short (≤5 min) | 1.00 | | 1.00 | | 1.00 | |
| Short (>5–10 min) | 0.96 | 0.66–1.39 | 0.76 | 0.48–1.20 | 0.83 | 0.50–1.39 |
| Medium (>10–20 min) | 1.13 | 0.77–1.67 | 0.87 | 0.55–1.3 | 1.13 | 0.66–1.93 |
| Long (>20–30 min) | 1.36 | 0.79–2.32 | 0.92 | 0.49–1.74 | 2.08 | 0.99–4.35 |
| Very long (>30 min) | 1.03 | 0.45–2.37 | 0.22 | 0.04–1.11 | 0.66 | 0.15–2.94 |
| Travel time to nearest practice of the patient's GP | | | | | | |
| Very short (≤5 min) | 1.00 | | 1.00 | | 1.00 | |
| Short (>5–10 min) | 1.04 | 0.79–1.36 | 1.10 | 0.80–1.52 | 1.18 | 0.81–1.71 |
| Medium (>10–20 min) | 1.17 | 0.83–1.65 | 0.92 | 0.60–1.42 | 1.14 | 0.70–1.83 |
| Long (>20–30 min) | 1.59 | 0.70–3.61 | 0.65 | 0.21–1.99 | 0.72 | 0.20–2.58 |
| Very long (>30 min) | 1.22 | 0.42–3.54 | 0.37 | 0.06–2.46 | 3.42 | 0.87–13.46 |
| Travel time to nearest outpatient service, either card. or non-card. | | | | | | |
| Very short (≤5 min) | 1.00 | | 1.00 | | 1.00 | |
| Short (>5–10 min) | 0.87 | 0.69–1.10 | 0.98 | 0.74–1.30 | 1.12 | 0.81–1.57 |
| Medium (>10–20 min) | 0.83 | 0.54–1.28 | 0.96 | 0.59–1.55 | 0.60 | 0.31–1.14 |
| Long (>20–30 min) | 1.10 | 0.49–2.49 | 0.87 | 0.25–3.01 | 1.61 | 0.51–5.03 |

[a] Adjusted for age, sex, length of stay, intensive care, discharge from cardiology, history of heart failure, diabetes, chronic kidney disease, previous use of ACEIs/ARBs, and general home care/HF care pathway/cardiology visit/use of first-line medications during follow-up.

[b] Adjusted for age, sex, length of stay, intensive care, discharge from cardiology, and general home care/HF care pathway/cardiology visit/use of first-line medications during follow-up.

[c] Adjusted for age, sex, length of stay, intensive care, discharge from cardiology, history of heart failure, cardiac arrhythmias, chronic kidney disease, dementia, previous use of diuretics/statins, and general home care/HF care pathway/cardiology visit/use of first-line medications during follow-up.

In addition to travel times, we analysed the degree of urbanisation of the community where the patient lived, a measure adopted in many other studies [15,16,18,34]. As expected, we found that travel times were longer for patients living in less densely populated areas. However, contrary to other studies that found poorer outcomes among rural residents [15,16,18,35,36], we found that rural patients had more follow-up cardiology visits. The better access to follow-up care by rural patients might be explained by a seamless organisation of care that is easier to implement in non-urban communities than in metropolitan settings, where the higher number of providers and facilities can make paradoxically more complex to refer patients to the same professionals. To support this, we found that rural patients were more commonly registered at the HF-CP, whose aim is to standardise care and to define the reference professional for the patient.

Still, no other significant associations between degree of urbanisation and outcomes were present in this study. Although urbanisation level is commonly seen as a proxy for SES and other SES-related characteristics [17], one possible explanation is that such inequalities are not unevenly distributed across the LHA of Bologna. Another possible explanation is that the

strong universal health coverage of Italy might play a role in reducing inequalities in access to healthcare, as suggested by a study evaluating the relationships between SES and HF outcomes in a universal national health service [37]. However, we lack updated information on census track-level SES in our catchment area, and the available data are not equipped to disentangle the association between geography and SES among patients with HF.

Because geographic location is also a proxy for the availability of appropriate healthcare facilities, some studies hypothesised that the worst outcomes of rural patients are due to lack of appropriate healthcare and to limited access to specialist services [16,17,35,38]. However, a strength of our study is that we adjusted all analyses for provision of HF-CP and inpatient/outpatient cardiologist care, which are known (and we found) to correlate with better outcomes [14,19,24,29,38,39]. Involvement of cardiologists in the management of patients with HF might reduce mortality and readmissions and enhance adherence to guideline treatments by endorsing or refining GP recommendations. Moreover, the collaboration with GPs and outpatient nurses can provide additional monitoring of the patients' concerns and adherence as planned in the CP [19,29,38]. Still, a shortcoming of our analysis is that the number of GP visits for each patient, either registered or not registered at the HF-CP, is not available. In Italy, most GPs use dedicated computer programmes to manage rosters, appointments and clinical data, but this information cannot be accessed by the local healthcare authorities.

Although rural patients had more follow-up cardiology visits ($X \rightarrow M$) and these visits were associated with lower mortality ($M \rightarrow Y$), we did not find evidence of reduced mortality among rural patients ($X \rightarrow Y$). This should come as no surprise because, as a causal process becomes more distal, the size of the effect typically gets smaller and ultimately fails to achieve statistical significance [40]. A possible interpretation is that the relationship between geographic location and mortality acts via a number of many other intermediary links, competing risks and random factors.

## Study limitations

As mentioned above, there are a number of limitations to our study. First, we lack some relevant information, including clinical features, disease severity, SES, informal caregivers, and GP visits. Second, we assumed that all patients would attend the nearest facility. Third, travel times do not fully depict the travel burden of patients with HF, who are elderly and can be hindered by a relevant combination of disabilities and clinical conditions.

## Conclusions

Our findings show that the travel times to healthcare services have no impact on the quality of care for patients with HF. Possible reasons for this result include the large number of healthcare services in the LHA of Bologna and the relatively short driving distances to the nearest facility (mostly <30 minutes), which can harm the generalisability of this study to rural areas with remote and isolated communities. We also found that the impact of the degree of urbanisation was possibly mediated by more proximal factors, some of which are potential targets for intervention such as the availability and utilisation of different types of care settings. More specifically, we found that cardiology services and multidisciplinary models of care had an impact on the quality of care for patients with HF.

When geographic accessibility is generally good, healthcare delivery and patient outcomes can be optimised by prioritising high-quality models of care, not only the quantity of available services being provided. This is particularly relevant considering the ageing of populations, the increase in disabilities and chronic diseases, and the downward trend of available healthcare resources for patients with complex needs.

Because the effectiveness of such composite programmes is context-specific, further research is needed to guide, tailor and improve healthcare settings and interventions to manage the clinical complexity and frailty of patients with HF.

## Supporting information

**S1 Dataset. Supporting data.**
(XLS)

**S1 Fig. Maps of the Local Healthcare Authority of Bologna, Northern Italy.** The shapefiles used to make this figure are made publicly available by the Italian National Institute of Statistics. Administrative boundaries reproduced from [31] under a CC BY license, with permission from the Italian National Institute of Statistics, original copyright 2019.
(PDF)

**S2 Fig. Diagram depicting selection of the study population.** GP, general practitioner.
(PDF)

**S1 Table. Description of data sources.**
(PDF)

**S2 Table. Six-month study outcomes, degree of urbanisation and travel times to healthcare.** ER, emergency room; GP, general practitioner; FUP, follow-up.
(PDF)

**S3 Table. Comorbidities and drug therapies considered for inclusion in multivariable regression models.**
(PDF)

**S4 Table. Distribution of comorbidities and previous medication use in the study population.**
(PDF)

**S5 Table. Association of processes of care, urbanisation levels and travel times with the study outcomes of patients with heart failure.**
(PDF)

## Acknowledgments

We wish to thank Maria Cristina Pirazzini, MS, of the LHA of Bologna for assistance in interpreting the data, and Professor Mark A. Hlatky, MD, of Stanford University for very helpful discussions and criticism.

## Author Contributions

**Conceptualization:** Jacopo Lenzi, Vera Maria Avaldi.

**Data curation:** Dario Molinazzi, Veronica Cappelli.

**Formal analysis:** Jacopo Lenzi.

**Investigation:** Jacopo Lenzi, Vera Maria Avaldi, Veronica Cappelli.

**Methodology:** Jacopo Lenzi, Vera Maria Avaldi.

**Project administration:** Maria Pia Fantini.

**Supervision:** Carlo Descovich, Stefano Urbinati.

**Validation:** Dario Molinazzi.

**Visualization:** Jacopo Lenzi.

**Writing – original draft:** Jacopo Lenzi, Vera Maria Avaldi.

**Writing – review & editing:** Jacopo Lenzi, Carlo Descovich, Stefano Urbinati, Maria Pia Fantini.

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
