## [Decision Letter · Decision Letter 0]

21 Aug 2019

PONE-D-19-20409

Are degree of urbanisation and travel times to healthcare services associated with the processes of care and outcomes of heart failure? A retrospective cohort study based on administrative data

PLOS ONE

Dear Dr. Avaldi,

Thank you for submitting your manuscript to PLOS ONE. After careful consideration, we feel that it has merit but does not fully meet PLOS ONE’s publication criteria as it currently stands. Therefore, we invite you to submit a revised version of the manuscript that addresses the points raised during the review process.

ACADEMIC EDITOR: 

In addition to the points raised by the reviewers, I would like you to address the following points:

1) It is acceptable that not all information on the methods is provided in the text if previously published. Still, the readers must be able to understand the methods without reading the cited paper. This means that that a short description of the according parts of the methods is required. Only citing a previous study is not sufficient.

2) The authors include some limitations of their analysis but lack to mention all of them. The reviewers addressed some of them and they also made suggestions that can possibly not be addressed based on the administrative data set. The authors must clearly address this. In this regard, a very important shortcoming is that there is no information on the severity of HF. 

3) Instead of repeating findings previously mentioned as part of the discussion / conclusion, the authors should focus more on the reasons why they think that their findings are in some contrast to other findings and possibly also their expectations (a clear hypothesis would help in this regard), more thoughts about the potential clinical implications and future perspectives, considering expected changes in healthcare in the future (as in part also addressed by the reviewers). If the authors do not want to extend too much regarding length of the text, the introduction can be shortened. This particularly refers to the extensive discussion about the burden of heart failure. This is well known, mentioned in thousands of papers, and can therefore be shortened significantly.

4) Please check that the references to tables and figures are correct. Suppl. text 1 is a figure. Please check that the manuscript follows the requirements of the journal regarding format, placement of tables, figures, references etc.

We would appreciate receiving your revised manuscript by Oct 05 2019 11:59PM. To enhance the reproducibility of your results, we recommend that if applicable you deposit your laboratory protocols in protocols.io, where a protocol can be assigned its own identifier (DOI) such that it can be cited independently in the future. For instructions see: http://journals.plos.org/plosone/s/submission-guidelines#loc-laboratory-protocols

We look forward to receiving your revised manuscript.

Kind regards,

Hans-Peter Brunner-La Rocca, M.D.

Academic Editor

PLOS ONE

Journal Requirements:

1. Given the retrospective design, the study can only draw conclusions about an association, please revise the language in the manuscript (and abstract) which refers to an ‘impact’ (or lack of ‘Impact’) of urbanisation and travel times to healthcare services on outcomes of heart failure or similar causal language, to refer to an association.

Reviewers' comments:

Reviewer's Responses to Questions

**Comments to the Author**

1. Is the manuscript technically sound, and do the data support the conclusions?

Reviewer #1: Yes

Reviewer #2: Yes

2. Has the statistical analysis been performed appropriately and rigorously? 

Reviewer #1: Yes

Reviewer #2: Yes

3. Have the authors made all data underlying the findings in their manuscript fully available?

Reviewer #1: Yes

Reviewer #2: Yes

4. Is the manuscript presented in an intelligible fashion and written in standard English?

Reviewer #1: Yes

Reviewer #2: Yes

5. Review Comments to the Author

Reviewer #1: Dear Author,

Many thanks for the opportunity to review this interesting manuscript examining the relationship between urbanisation and travel time with processes of care and heart failure outcomes. In general it was well written and structured with transparency and rigour from data to conclusion. I would like to make few comments which I would like your consideration.

In terms of participant characteristics, it would have been beneficial to know NYHA class &/or the EF. This was particularly pertinent as on review of Table 1, 18 (4.3%) of rural compared to 27 (2.8) city and 22 (3.4) towns & suburbs were discharged from intensive care. Was there a difference in the "sickness" of the patients"?

Figure 1 demonstrated a numerous GP practices and centralised cardiology wards. Can I ask why GP visits were not recorded or included as a study outcome?

Use of Doppler echocardiogram during hospital stay noted a potential confounder- why? Surely all HF patients will have an objective assessment.

Results were displayed appropriately in table with main point summarised in text. Please amend Table 4 as "degree of urbanisation" data is repeated. I am unsure of the value of SI table. Also please refer to S1 figure in main text to ensure reader knows to access the information. It is important to ensure clear land marking of the supplementary material.

Results inform conclusions that travel time and urbanisation had no effect on processes of care and outcomes. Cardiology visits were more frequent among rural patients. It would have been interesting to hear your thoughts on how this will change in the future in light of declining health resources and a growing elderly population with multiple comorbidities.

Reviewer #2: Dear authors

Please find my remarks / suggestions on your publication 'Are degree of urbanisation and travel times to healthcare services associated with the processes of care and outcomes of heart failure? A retrospective cohort study based on administrative data'.

General remark: authors assume that readers have knowledge how local health care is organised. Despite they tried to optimally explain the organisation, it is not always evident to understand the organisation. Some questions reflect on this topic.

Sentence 63-65: Would the authors please rephrase: 'However, these studies focused mainly on patients with cancer and were carried out using very different data sources and methods, therefore more research is needed to obtain further evidence on this topic.'

Sentences 71-72: 'Also, there is a lack of evidence regarding transport accessibility measured as the actual distance between the patient's home and the healthcare facility.'

Sentences 177-180: Potential confounders: In order to minimise the potential confounding of individual characteristics on the association of urbanisation level and travel times with outcomes, we retrieved some patient baseline characteristics.

These included:

1. Demographic characteristics (age, sex and citizenship)

Sentences 388-343: In the discussion authors discussed the travel burden for elderly patients, and the fact that elderly patients often must be accompanied by their caregivers...

Reviewer: Accessibility of transport also means the easiness of getting transport. In case of elderly patients the presence of informal caregivers might be of utmost importance. The authors included 'citizenship' into the confounders. Yet, citizenship does not cover the presence / availability of informal caregivers to transport patients to a health care facility... Authors discuss the importance of family members / informal caregivers in relation to the transport.

I would like to ask the authors to add ‘presence of informal caregivers’ to the confounders and investigate / show the results of this important aspect.

Do the authors have information about the ratio of no-shows in the several urbanisation degrees. Do patients of the city less frequently have a no-show visit compared to rural patients? If patients do not visit health care facilities (due to transport issues, for example due to lacking transport support), this might be considered as a bad result, which now is not visible in the results.

Table 4 contains double information about 'degree of urbanization': please remove double information.

Sentence 349-355: Authors discuss about HF-CP and the fact that it is easier to organize HF-CP in rural area. Authors show the health care services in a map: S1 Text. Maps of the Local Healthcare Authority of Bologna, Northern Italy. For me as a reader it is not clear where HF-CP are located. The map shows several services, yet no HF-CP. Please add where HF-CP are organised.

Rural patients are receiving more follow-up visits from a cardiologist. Are cardiologists always functioning into outpatient cardiology services or do they have also consulting hours in other services? Or involved into the HF-CP? Please add this information.

Authors investigated a very local region. Do the authors have advice for other regions?

6. PLOS authors have the option to publish the peer review history of their article (what does this mean?). If published, this will include your full peer review and any attached files.

Reviewer #1: No

Reviewer #2: No

---

## [Author Response · Author response to Decision Letter 0]

9 Sep 2019

ACADEMIC EDITOR

We thank the Editor for the helpful comments.

1) It is acceptable that not all information on the methods is provided in the text if previously published. Still, the readers must be able to understand the methods without reading the cited paper. This means that that a short description of the according parts of the methods is required. Only citing a previous study is not sufficient.

Reply: We thank the Editor for his suggestion. We have now provided a more accurate description of the statistical methods used for confounder selection (Ln 229-34).

2) The authors include some limitations of their analysis but lack to mention all of them. The reviewers addressed some of them and they also made suggestions that can possibly not be addressed based on the administrative data set. The authors must clearly address this. In this regard, a very important shortcoming is that there is no information on the severity of HF.

Reply: We have now mentioned among the study limitations the absence of information on HF severity and other relevant information, including GP visits and presence of caregivers (Ln 334-7, 350-1, 381-5).

3) Instead of repeating findings previously mentioned as part of the discussion / conclusion, the authors should focus more on the reasons why they think that their findings are in some contrast to other findings and possibly also their expectations (a clear hypothesis would help in this regard), more thoughts about the potential clinical implications and future perspectives, considering expected changes in healthcare in the future (as in part also addressed by the reviewers). If the authors do not want to extend too much regarding length of the text, the introduction can be shortened. This particularly refers to the extensive discussion about the burden of heart failure. This is well known, mentioned in thousands of papers, and can therefore be shortened significantly.

Reply: We thank the Editor for his thoughtful suggestion. We have enhanced the conclusions to highlight the practical implications of our findings, as well as the future perspective related to health resource decline and population ageing (Ln 404-8). In particular, we argued that healthcare delivery could be optimised by prioritising quality, not only quantity of services being provided. The contrast with other findings is addressed in Ln 364-9, 374-7 and 396-8, where we discuss SES distribution, universal health coverage, multidisciplinary interventions and relatively short driving distances. Lastly, as suggested, the introduction has been shortened (Ln 49-52).

4) Please check that the references to tables and figures are correct. Suppl. text 1 is a figure. Please check that the manuscript follows the requirements of the journal regarding format, placement of tables, figures, references etc

Reply: Supplementary files have been renamed according to their content. We have also double-checked all formatting guidelines, including references and figure requirements.

REVIEWER #1

Many thanks for the opportunity to review this interesting manuscript examining the relationship between urbanisation and travel time with processes of care and heart failure outcomes. In general it was well written and structured with transparency and rigour from data to conclusion. I would like to make few comments which I would like your consideration.

Reply: We thank the reviewer for this positive comment. We hope that we have addressed adequately each of the issues she/he raised.

In terms of participant characteristics, it would have been beneficial to know NYHA class &/or the EF. This was particularly pertinent as on review of Table 1, 18 (4.3%) of rural compared to 27 (2.8) city and 22 (3.4) towns & suburbs were discharged from intensive care. Was there a difference in the "sickness" of the patients"?

Reply: Unfortunately, information on NYHA class and ejection fraction is not routinely collected in the Italian hospital discharge records. We have now mentioned this major shortcoming in the discussion (Ln 334-7).

Figure 1 demonstrated a numerous GP practices and centralised cardiology wards. Can I ask why GP visits were not recorded or included as a study outcome?

Reply: We thank the reviewer for raising this point. In Italy, most GPs use dedicated computer programmes to manage their rosters, appointments and clinical data. However, these programmes cannot be accessed by the Local Healthcare Authorities. The lack of information on GP visits has been now addressed in the discussion (Ln 381-5).

Use of Doppler echocardiogram during hospital stay noted a potential confounder- why? Surely all HF patients will have an objective assessment.

Reply: We thank the reviewer for her/his remark. Initially we included Doppler echocardiography in the analysis, but then decided to discard it because most patients with heart failure get this evaluation, as the reviewer correctly pointed out. Unfortunately, we forgot to erase this information from our previous versions of the manuscript. The text has been amended accordingly (Ln 178-9 and table footnotes). 

Results were displayed appropriately in table with main point summarised in text. Please amend Table 4 as "degree of urbanisaton" data is repeated. I am unsure of the value of SI table. Also please refer to S1 figure in main text to ensure reader knows to access the information. It is important to ensure clear land marking of the supplementary material.

Reply: We thank the reviewer for noticing these inaccuracies—Table 4 has been corrected, and supplementary files have been renamed according to their content. We think that S1 Table might be of some value for those who are not familiar with multivariate statistics, that is, the simultaneous analysis of multiple outcomes. Because making ourselves clear is our priority, we would rather keep this synthesis matrix as supporting information.

Results inform conclusions that travel time and urbanisation had no effect on processes of care and outcomes. Cardiology visits were more frequent among rural patients. It would have been interesting to hear your thoughts on how this will change in the future in light of declining health resources and a growing elderly population with multiple comorbidities.

Reply: We thank the reviewer for her/his suggestion. We have enhanced the conclusions to highlight the practical implications of our findings, as well as the future perspective related to health resource decline and population ageing (Ln 404-8). In particular, we argued that healthcare delivery could be optimised by prioritising quality, not only quantity of services being provided.

REVIEWER #2

General remark: authors assume that readers have knowledge how local health care is organised. Despite they tried to optimally explain the organisation, it is not always evident to understand the organisation. Some questions reflect on this topic.

Reply: We thank the reviewer for the helpful comments. We hope that we have better clarified how healthcare services are organised in our Local Healthcare Authority.

Sentence 63-65: Would the authors please rephrase: 'However, these studies focused mainly on patients with cancer and were carried out using very different data sources and methods, therefore more research is needed to obtain further evidence on this topic.'

Reply: The sentence has been rephrased as follows: “However, this review included mainly cancer research studies relying on different data sources and variables” (Ln 58-9).

Sentences 71-72: 'Also, there is a lack of evidence regarding transport accessibility measured as the actual distance between the patient's home and the healthcare facility.'

Reply: The sentence has been rephrased as follows: “A potential limitation of these studies is the lack of information regarding transport accessibility measured as the actual distance between the patient’s home and the healthcare facility” (Ln 64-6).

Sentences 177-180: Potential confounders: In order to minimise the potential confounding of individual characteristics on the association of urbanisation level and travel times with outcomes, we retrieved some patient baseline characteristics. These included:

1. Demographic characteristics (age, sex and citizenship)

Reply: The sentence has been rephrased as follows: “We collected some patient baseline characteristics to reduce the potential source of confounding. These included: 1) Age; 2) Sex; 3) Citizenship” (Ln 172-6).

Sentences 388-343: In the discussion authors discussed the travel burden for elderly patients, and the fact that elderly patients often must be accompanied by their caregivers...

Reviewer: Accessibility of transport also means the easiness of getting transport. In case of elderly patients the presence of informal caregivers might be of utmost importance. The authors included 'citizenship' into the confounders. Yet, citizenship does not cover the presence / availability of informal caregivers to transport patients to a health care facility... Authors discuss the importance of family members / informal caregivers in relation to the transport.

I would like to ask the authors to add ‘presence of informal caregivers’ to the confounders and investigate / show the results of this important aspect.

Reply: We thank the reviewer for raising this important point. Unfortunately, the presence of informal caregivers is not reported in the Italian hospital discharge records. To the best of our knowledge, administrative data sources of many other countries also lack this information. We have now mentioned this major shortcoming in the discussion (Ln 350-1).

Do the authors have information about the ratio of no-shows in the several urbanisation degrees. Do patients of the city less frequently have a no-show visit compared to rural patients? If patients do not visit health care facilities (due to transport issues, for example due to lacking transport support), this might be considered as a bad result, which now is not visible in the results.

Reply: Unfortunately, this information is not available because our administrative databases include only visits that are booked AND provided to the patient. In Emilia-Romagna there is a regional law that should discourage no-shows, because patients that do not show up without cancellation have to pay the full patient contribution (Legge regionale 2/2016: “Norme regionali in materia di organizzazione degli esercizi farmaceutici e di prenotazioni di prestazioni specialistiche ambulatoriali” art. 23 comma 3). For this reason, it is reasonable to assume that the number of no-shows is limited in the catchment area of Bologna.

Table 4 contains double information about 'degree of urbanization': please remove double information.

Reply: We thank the reviewer for noticing the inaccuracy—Table 4 has been corrected.

Sentence 349-355: Authors discuss about HF-CP and the fact that it is easier to organize HF-CP in rural area. Authors show the health care services in a map: S1 Text. Maps of the Local Healthcare Authority of Bologna, Northern Italy. For me as a reader it is not clear where HF-CP are located. The map shows several services, yet no HF-CP. Please add where HF-CP are organised.

Reply: The HF-CP is a structured multidisciplinary care plan that promotes integration between primary and secondary care, and details essential steps in the care of patients. The GP remains the gatekeeper for the patients and coordinates with cardiologists and nurses for an easier access to consultation and counselling to improve lifestyle and optimise medication adherence. There are no specific HF-CP centres or facilities in the catchment area, because registered patients access GP practices, cardiology services or ambulatory care nursing practices in case they need medical care or counselling. All these aspects have been clarified in the Methods section (Ln 194-203).

Rural patients are receiving more follow-up visits from a cardiologist. Are cardiologists always functioning into outpatient cardiology services or do they have also consulting hours in other services? Or involved into the HF-CP? Please add this information.

Reply: We have now stated in the Methods section that cardiology follow-up visits can be provided either in inpatient cardiology services or in outpatient cardiology services (Ln 125). Cardiologists in either settings of care can see all patients with HF.

Authors investigated a very local region. Do the authors have advice for other regions?

Reply: We thank the reviewer for her/his question. Our findings cannot be generalised to isolated and remote communities, but we gave some advice for other regions where driving distances to healthcare facilities are relatively short. In particular, we argued that healthcare delivery could be optimised by prioritising quality, not only quantity of services being provided (Ln 404-8).

---

## [Decision Letter · Decision Letter 1]

23 Sep 2019

PONE-D-19-20409R1

Are degree of urbanisation and travel times to healthcare services associated with the processes of care and outcomes of heart failure? A retrospective cohort study based on administrative data

PLOS ONE

Dear Dr. Avaldi,

Thank you for submitting your manuscript to PLOS ONE. After careful consideration, we feel that it has merit but does not fully meet PLOS ONE’s publication criteria as it currently stands. Therefore, we invite you to submit a revised version of the manuscript that addresses the points raised during the review process.

The manuscript has significantly improved. However as suggested by both reviewers, there are minor aspects remaining which I would like you to address before your manuscript is ready for publication. Please note that PLOS ONE does not type edit accepted manuscript. This requires that manuscripts need to be in standard English and even small adjustments may be required prior to acceptance of a manuscript.

We would appreciate receiving your revised manuscript by Nov 07 2019 11:59PM. To enhance the reproducibility of your results, we recommend that if applicable you deposit your laboratory protocols in protocols.io, where a protocol can be assigned its own identifier (DOI) such that it can be cited independently in the future. For instructions see: http://journals.plos.org/plosone/s/submission-guidelines#loc-laboratory-protocols

We look forward to receiving your revised manuscript.

Kind regards,

Hans-Peter Brunner-La Rocca, M.D.

Academic Editor

PLOS ONE

Reviewers' comments:

Reviewer's Responses to Questions

**Comments to the Author**

1. If the authors have adequately addressed your comments raised in a previous round of review and you feel that this manuscript is now acceptable for publication, you may indicate that here to bypass the “Comments to the Author” section, enter your conflict of interest statement in the “Confidential to Editor” section, and submit your "Accept" recommendation.

Reviewer #1: All comments have been addressed

Reviewer #2: All comments have been addressed

2. Is the manuscript technically sound, and do the data support the conclusions?

Reviewer #1: Yes

Reviewer #2: Yes

3. Has the statistical analysis been performed appropriately and rigorously? 

Reviewer #1: Yes

Reviewer #2: Yes

4. Have the authors made all data underlying the findings in their manuscript fully available?

Reviewer #1: Yes

Reviewer #2: Yes

5. Is the manuscript presented in an intelligible fashion and written in standard English?

Reviewer #1: Yes

Reviewer #2: Yes

6. Review Comments to the Author

Reviewer #1: Dear Author,

Many thanks for the opportunity to review your revised manuscript. You have clearly and concisely addressed initial concerns, which have strengthened your submission.

Please consider the following comments which remain outstanding

Sentence 65: Please remove repetition of the words " relying on different"

Sentence 77: Change the word "founded" to "established"

Sentence 83: Remove the word "However" as not required

Sentence 86: Ensure consistency - consider rephrasing to " The objectives of this study were to investigate whether urbanisation levels and travel time to healthcare services in the LHA of Bologna, are associated with processes of care and outcomes of patients with heart failure".

Sentence 100: Could you provide details of the data sources, perhaps in a table format?

Sentence 180: Insert the word "variable" after confounding

Sentence 212: Please rephrase

Sentence 257: Consider using the term pseudonymised instead of "de-identified"

Sentence 330- Clarify there are indeed 2 tables "A" and "B" within S4

There are a number of limitations noted within the discussion section. It might be more appropriate to group these together into one paragraph, titled limitations.

Best wishes as you proceed with this submission

Reviewer #2: Thanks for your answers and for adapting the manuscript according to the comments.

I've found one sentence which has to be corected: Sentence 58-59: double text:

However, this review included mainly cancer research studies relying on different relying on different data sources and variables.

7. PLOS authors have the option to publish the peer review history of their article (what does this mean?). If published, this will include your full peer review and any attached files.

Reviewer #1: No

Reviewer #2: No

---

## [Author Response · Author response to Decision Letter 1]

25 Sep 2019

REVIEWER #1

Sentence 65: Please remove repetition of the words "relying on different"

Reply: We thank the reviewer for noticing this inaccuracy. The text has been amended.

Sentence 77: Change the word "founded" to "established"

Reply: The verb has been changed as suggested.

Sentence 83: Remove the word "However" as not required

Reply: We agree with the reviewer that the two paragraphs are congruent and no adverb is required to link them. The text has been amended.

Sentence 86: Ensure consistency - consider rephrasing to "The objectives of this study were to investigate whether urbanisation levels and travel time to healthcare services in the LHA of Bologna, are associated with processes of care and outcomes of patients with heart failure".

Reply: We thank the reviewer for raising this point. Our introduction is “travel time-oriented”, that is, we highlight that there is lack of evidence regarding transport accessibility measured as the actual distance between the patient’s home and the healthcare facility. We organised our study aim on purpose to give priority to travel times, and added that urbanisation was also assessed to be consistent with existing literature. We really appreciate the reviewer’s suggestion, but we feel that putting urbanisation level and travel times on the same level in the last paragraph would make the whole introduction section less flowing.

Sentence 100: Could you provide details of the data sources, perhaps in a table format?

Reply: A new supplementary table (S1) has been included, as suggested.

Sentence 180: Insert the word "variable" after confounding

Reply: In this sentence, “confounding” is a noun. A number of occurrences where “confounding” is used as a noun can be borrowed from the literature (e.g., Occup Environ Med 2003;60:227-34).

Sentence 212: Please rephrase

Reply: We have rephrased the sentence as follows: “There are no facilities specifically dedicated to the HF-CP: registered patients can access GP practices, cardiology services or ambulatory care nursing practices in case they need medical care or counselling.”

Sentence 257: Consider using the term pseudonymised instead of "de-identified"

Reply: We thank the reviewer for her or his suggestion. The verb has been changed as suggested.

Sentence 330: Clarify there are indeed 2 tables "A" and "B" within S4

Reply: This aspect has been clarified for S3, S4 and S5 Tables.

There are a number of limitations noted within the discussion section. It might be more appropriate to group these together into one paragraph, titled limitations.

Reply: We agree with the reviewer that grouping together all limitations in a single paragraph might be desirable. The journal gives a degree of freedom to the authors, who are not forced to fill prearranged sections when writing the papers. We took advantage of this to write a discussion where methodological pitfalls are in the focus. The reviewer can see that a number of important limitations, such as the lack of relevant clinical information, are at the very beginning of the discussion—the editor asked us to give much relevance to such limitations. For this reason, we have kept the discussion unchanged, but added a subsection where all limitations are now summarised.

REVIEWER #2

I've found one sentence which has to be corrected: Sentence 58-59: double text: “However, this review included mainly cancer research studies relying on different relying on different data sources and variables.”

Reply: We thank the reviewer for noticing this inaccuracy. The text has been amended.

---

## [Editor Report · Decision Letter 2]

1 Oct 2019

Are degree of urbanisation and travel times to healthcare services associated with the processes of care and outcomes of heart failure? A retrospective cohort study based on administrative data

PONE-D-19-20409R2

Dear Dr. Avaldi,

We are pleased to inform you that your manuscript has been judged scientifically suitable for publication and will be formally accepted for publication once it complies with all outstanding technical requirements.

With kind regards,

Hans-Peter Brunner-La Rocca, M.D.

Academic Editor

PLOS ONE

---

## [Editor Report · Acceptance letter]

21 Oct 2019

PONE-D-19-20409R2 

Are degree of urbanisation and travel times to healthcare services associated with the processes of care and outcomes of heart failure? A retrospective cohort study based on administrative data 

Dear Dr. Avaldi:

I am pleased to inform you that your manuscript has been deemed suitable for publication in PLOS ONE. Congratulations! Your manuscript is now with our production department. 

With kind regards,

on behalf of

Dr. Hans-Peter Brunner-La Rocca 

Academic Editor

PLOS ONE